# Attenuated Amplitude of Pattern Electroretinogram in Glaucoma Patients with Choroidal Parapapillary Microvasculature Dropout

**DOI:** 10.3390/jcm11092478

**Published:** 2022-04-28

**Authors:** Jiyun Lee, Chan Kee Park, Kyoung In Jung

**Affiliations:** Department of Ophthalmology, Seoul St. Mary’s Hospital, College of Medicine, The Catholic University of Korea, Seoul 06591, Korea; lemonssung@hanmail.net (J.L.); ckpark@catholic.ac.kr (C.K.P.)

**Keywords:** microvascular dropout, β-zone parapapillary atrophy, pattern electroretinogram, optical coherence tomography angiography, N95 amplitude, glaucoma

## Abstract

This study aims to investigate whether parapapillary choroidal microvasculature dropout (MvD) is related to visual function measured by pattern electroretinogram (PERG) in glaucomatous eyes with β-zone parapapillary atrophy (PPA). A total of 79 patients with open angle glaucoma and preperimetric glaucoma with β-zone PPA was included in this cross-sectional study. Through the deep layer of the Swept-source optical coherence tomography angiography image, the angular width and the area of MvD were measured. Visual function was evaluated with a standard automated perimetry and PERG. N95 and P50 PERG amplitudes in eyes with MvD were noticeably decreased compared to those without MvD (*p* = 0.004 and *p* = 0.007, respectively), although the mean deviation was not significantly different (*p* = 0.107). The lower N95 amplitude was associated with the presence of MvD (β = −0.668, *p* = 0.017) and wider angular width of MvD (B = −7.612, *p* = 0.014). Old age (*p* = 0.001), average ganglion cell’s inner plexiform layer thickness (*p* = 0.003), and the presence of MvD (*p* = 0.020) were significantly related to low N95 amplitude. Association between the presence and extent of the MvD and PERG amplitudes suggests that the presence of MvD has relevance to the generalized dysfunction of retinal ganglion cells.

## 1. Introduction

Glaucoma diagnosis is mostly based on “structural tests” (stereophotographs [1,2], optical coherence tomography (OCT)) [3,4], and “functional tests” (visual field (VF)) [5,6,7]. However, when it comes to detecting the development of glaucoma from glaucoma suspects [8,9,10] or predicting disease progression in patients with advanced stage [11,12,13,14], due to discrepancies between tests, these conventional tests have limitations. OCT angiography (OCTA) [15,16,17,18] and pattern electroretinogram (PERG) [19,20,21] could be used as ancillary methods. 

Through OCTA, the vascular flow of various layers in the optic nerve head (ONH) and macular was investigated [22,23,24,25]. Moreover, parapapillary choroidal microvascular dropout (MvD), a focal sectoral lesion with completely compromised choriocapillaris in the deep layer, was found [26,27,28,29,30]. Although its pathogenesis remains unclear, many studies have found that MvD could be an indicator of compromised ocular or systemic perfusion in glaucoma patients. In addition, MvD was reported to be related to structural deformation around the optic nerve head such as focal lamina defect and parapapillary scleral deformation [27,31,32]. Based on these findings, MvD might have an impact on glaucoma progression [33,34].

Regarding PERG, it mainly measures the electrical activity of the inner retina including retinal ganglion cells (RGC) [35]. According to previous studies [36,37,38], compared to the OCT and VF tests, PERG outperformed both in identifying early dysfunctional but still viable RGCs. Given the strength of PERG, glaucomatous eyes with attenuated PERG parameters are more likely to have a substantial portion of sick RGCs, escalating under predominant mechanical and vascular stress.

Based on recently reported findings using OCTA and PERG [39,40], we hypothesized that patients accompanied with MvD might show declined visual function in PERG. In this study, we evaluated whether the presence of MvD or quantitative parameters of MvD was associated with the amplitude of PERG. 

## 2. Materials and Methods

This cross-sectional study was approved by the Institutional Review Board (IRB) of the Catholic University of Korea, Seoul, Korea (approval number: KC21RISI0018). It was performed following the tenets of the Declaration of Helsinki. The requirement of informed consent was waived by the IRB of Seoul St. Mary’s Hospital, Korea, due to the retrospective design. 

### 2.1. Subjects

Data of patients who visited the glaucoma clinic at Seoul St. Mary’s Hospital from January 2017 to November 2020 were collected using electronic medical records. Patients were included when they met the following inclusion criteria: a best-corrected visual acuity (BCVA), ≥20/40, an open angle glaucoma or preperimetric glaucoma with PPA, and OCTA images with quality scores greater than 60. Exclusion criteria were: poor OCT images (due to misalignment, artifacts, and signal strength < 6), a history of retinal disease including diabetic and hypertensive retinopathy, a history of eye surgery except cataract operation, a history of a systemic or neurological disease that might affect VF, or axial length longer than 26.5 mm to minimize the impact of myopia in optic disc change. If both eyes were eligible, one eye per individual was selected on a random basis.

### 2.2. Measurements

Each participant underwent a full ophthalmic assessment, including measurement of BCVA, slit-lamp examination, Goldmann applanation tonometry, gonioscopy, central corneal thickness using pachymetry (Tomey Corporation, Nagoya, Japan), determination of axial length (AL) using ocular biometry (IOL Master; Carl Zeiss Meditec, Dublin, CA, USA), dilated stereoscopic examination of the optic disc and fundus, color disc photography, red-free RNFL photography (Canon, Tokyo, Japan), optical coherence tomography (OCT) (Cirrus OCT using software version 6.0; Carl Zeiss Meditec, Dublin, CA, USA), and Humphrey VF examination (24-2 Swedish Interactive Threshold Algorithm Standard program; Carl Zeiss Meditec, Dublin, CA, USA).

Open-angle glaucoma diagnosis was defined based on glaucomatous optic disc appearance (such as diffuse or localized rim thinning, a notch in the rim, or a cup-to-disc ratio higher than that of the other eye by >0.2), VF consistent with glaucoma (a cluster of ≥3 non-edge points on the pattern deviation plot with a probability of <5% of the normal population, with one of these points having a probability of <1%, a pattern standard deviation with *p* < 5%, or a Glaucoma Hemifield Test result outside the normal limits in a consistent pattern on two qualifying VFs) confirmed by two glaucoma specialists (K.I.J and C.K.P.), and an open-angle on gonioscopy. Preperimetric glaucoma was defined with normal visual field accompanied with either glaucomatous optic disc appearances (vertical cup-to-disc ratio higher than 0.5 or asymmetricity between two eyes more than 0.2) or preexisting RNFL defects.

### 2.3. Optical Coherence Tomography Angiography

The microvasculature of the peripapillary area was imaged via a swept-source OCTA device (DRI OCT Triton; Topcon, Tokyo, Japan) using a laser with a wavelength of 1050 nm and a speed of 100,000 A-scans per second. The OCTA provided En Face images through automated layer segmentation around the optic nerve head in four layers. Among those, the radial peripapillary capillary (RPC) mode was chosen to estimate a 70 μm thick layer below the internal limiting membrane (ILM) to calculate superficial parapapillary VD. To measure superficial macular VD, images of the superficial macular layer lying from 2.6 μm below the ILM to 15.6 μm below the junction of the inner plexiform layer and IPL/INL were obtained. To measure VDs of both superficial parapapillary and macular layers, we used a published method [31,41,42]. Images with a quality score less than 60 and unclear ocular vascular structures were excluded from further analysis.

### 2.4. Measurement of Parapapillary Choroidal Microvasculature Dropout

To assess MvD, the image of the deep layer was retrieved by manually resetting the starting point as retinal pigment epithelium (RPE) while keeping the original endpoint 390 μm below Bruch’s membrane. Since the original choroidal/disc mode was intended to measure from 130 μm below the ILM to 390 μm below Bruch’s membrane, the manual customization setting was arranged to avoid and minimize the effect of the superficial layer.

MvD was identified when the width of the dropout appeared to be greater than twice of that of visible juxtapapillary microvessels [43]. The area of the MvD was manually calculated using a tool integrated in the OCTA viewer software (IMAGEnet6; Topcon, Tokyo, Japan; Figure 1). Regarding large vessels across the MvD, we included the area covered by vessels on the condition that MvD existed beyond these vessels. Otherwise, the area occupied by large vessels was not counted.

In terms of the angular width of MvD, the same tool for measuring the area was applied. Two lines were drawn from the center of the optic disc to points where the disc margin and the border of MvD met. The angle was then measured. The area and angular width of the MvD were calculated by two authors (J.L and K.I.J) who were masked to clinical data. These values were determined as the mean of measurements performed by the two authors.

### 2.5. Electroretinography

An electroretinogram (ERG) stimulator (Neuro-ERG; Neurosoft, Ivanovo, Russia) that was commercially available was adopted. Before applying a total of four electrodes, participants with non-dilated pupils were placed in a dim room with a background illumination of 50. Two 35 mm Ag/AgCl skin electrodes were taped to the lower lids and two ground electrodes were taped to both earlobes. Black and white checkerboards with a check size of 1.81° were presented on a 24 inch-monitor with a 48° × 33° visual angle at a distance of 60 cm. Stimuli were modulated in counterphase at 4 Hz. These checkerboards had a mean luminance of 105 cd/m^2^. Participants fixed their views at the center of the monitor, where a red-colored fixed point was placed. The PERG was measured as binocular recordings with an appropriate refractive correction. Responses were band-pass filtered (1–50 Hz) and sampled at 10,000 Hz. At least 100 readings were recorded and averaged. To investigate the reproducibility of ERG parameters, test-retest variability was determined with 34 randomly selected measurements.

Latencies for N35, P50, and N95 were measured from the onset of checkerboard reversal to the peak of each component. P50 amplitude was estimated from the trough of N35 to the peak of P50. N95 amplitude was determined from the peak of P50 to the trough of N95. 

### 2.6. Statistical Analysis

All statistical analyses were performed using SPSS version 17.0 (SPSS, Inc., Chicago, IL, USA). Student’s *t*-test was adopted to compare differences between groups. Chi-square test was applied to compare frequencies. A *p*-value of less than 0.05 was considered statistically significant. Logistic regression analysis was used to ascertain factors associated with the presence of MvD. Linear regression analyses were applied to explore meaningful factors affecting either angular width of MvD or amplitudes of PERG. Variables with significance at *p* < 0.10 in univariate analysis were included in the multivariate model. *p* < 0.05 was considered to represent statistical significance.

## 3. Results

A total of 96 eyes of 96 OAG patients were initially recruited. Due to the absence of β-PPA, 17 patients were excluded from further analysis. The remaining 79 eyes with β-PPA were analyzed, including 46 (58.2%) eyes with MvD and 36 (41.8%) eyes without. The characteristics of patients are shown in Table 1. Based on the Hodapp-Anderson-Parish criteria [44], glaucoma patients ranging from preperimetric glaucoma to early or moderate stage were included (VF mean deviation (MD): −4.64 ± 6.53 dB). 

Comparison between eyes with MvD and those without MvD was performed. Results are shown in Table 2. In terms of glaucoma severity, there was no significant difference in MD, average RNFL thickness, or average ganglion cell inner plexiform layer (GCIPL) thickness. Moreover, VDs in parapapillary and macular regions were similar between the two groups. However, eyes with MvD had substantially decreased P50 and N95 amplitudes compared to eyes without MvD (*p* = 0.007 and *p* = 0.004, respectively). 

Logistic regression analysis was carried out to find factors associated with presence of MvD (Table 3). Declined amplitudes of P50 and N95 and macular vessel density were associated with the presence of MvD in univariate analysis. However, in multivariate analysis, only N95 amplitude was a significant parameter (β = −0.668, *p* = 0.017). 

Another linear regression analysis was performed to ascertain whether there was a correlation between factors and specific features of MvD such as angular width and area. According to the analysis for the correlation between angular width of MvD and parameters (Table 4), decreased N95 amplitude, decreased average RNFL thickness, and larger average cup to disc ratio were significantly linked to wider angular width (B = −7.612, *p* = 0.014; B = −0.796, *p* = 0.027; and B = 73.736, *p* = 0.065, respectively). Ultimately, in multivariate analysis, the N95 amplitude was substantially correlated with angular width of MvD (B = −7.612, *p* = 0.014; Appendix A). Meanwhile, unlike the angular width of MvD, no significant factor was associated with the area of MvD.

Table 5 shows outcomes of linear regression analysis identifying parameters associated with PERG amplitudes. Older age and decreased average GCIPL thickness were significantly associated with both P50 and N95 amplitudes (β = −0.026, *p* = 0.005; and β = 0.038, *p* = 0.007, respectively, in multivariate analysis for P50 amplitude; β = −0.047, *p* = 0.001 and β = 0.067, *p* = 0.003, respectively, in multivariate analysis for N95 amplitude). Although the presence of MvD showed a significant association with N95 amplitude (β = −0.815, *p* = 0.020), its association with P50 amplitude only showed a borderline significance (β = −0.428, *p* = 0.051).

Representative cases are shown in Figure 2 and Figure 3. Figure 2 compares an eye without MvD (Figure 2A–H) to an eye with MvD (Figure 2I–P). A 59-year-old female without MvD (Figure 2C) had an inferotemporal RNFL defect (Figure 2A,B). Her average RNFL and GCIPL thicknesses were 80 μm and 74 μm, respectively (Figure 2D–F). Her VF test showed a central VF defect with MD of −2.35 dB (Figure 2G). Her P50 and N95 amplitudes were 3.65 mV and 6.99 mV, respectively. The patient with MvD (Figure 2K) was a 60-year-old woman with an inferotemporal RNFL defect (Figure 2I,J). Her glaucoma severity was similar to the control regarding MD (−1.95 dB) (Figure 2O). Her average RNFL and GCIPL thicknesses were 85 μm and 70 μm, respectively (Figure 2L–N). Nonetheless, the measured PERG was lower than that of the control (2.66 mV for P50 amplitude and 5.6 mV for N95 amplitude; Figure 2P).

Another representative case demonstrating the association between the angular width of MvD and the N95 amplitude is shown in Figure 3. A 53-year-old male with multiple RNFL defects (Figure 3A,B) had a MvD with an estimated angular width of 9.9° (Figure 3C). His average RNFL and GCIPL thicknesses were 67 μm and 72 μm, respectively (Figure 3D–F). His VF MD was −8.43 dB (Figure 3G). His P50 and N95 amplitudes were 2.58 mV and 6.58 mV, respectively (Figure 3H). The other patient (Figure 3I,J) with a larger MvD (angular width 54.9°; Figure 3K) was a 63-year-old man. In spite of similar disease severity in terms of average RNFL and GCIPL thicknesses and MD (64 μm, 65 μm, and −8.92 dB, respectively; Figure 3L–O), PERG showed a decrease in the P50 amplitude and a substantial plunge in N95 amplitude (2.46 mV and 2.96 mV, respectively; Figure 3P). 

## 4. Discussion 

In this study, we found that in patients with PPA and similar degree of VF damage, eyes with MvD showed significantly lower N95 amplitudes than those without MvD (*p* = 0.004). The presence of MvD and thinner average GCIPL thickness were associated with lower N95 amplitude in multivariate linear regression analysis (*p* = 0.003 and *p* = 0.020, respectively). There was a negative correlation between the angular width of MvD and N95 amplitude (*p* = 0.014). 

To the best of our knowledge, this is the first study that investigates the relationship between parapapillary deep microvasculature and PERG parameters. We assumed that the presence of MvD might signify substantial decrease or dysfunction of RGCs, thereby giving rise to reductions of PERG amplitudes in glaucoma patients. Similar to our hypothesis, the existence of MvD has been regarded as a sign of worsening glaucoma in precedent studies [33,34,43,45]. 

Our group has demonstrated that PERG changes in preperimetric glaucoma or early-stage glaucoma earlier than 24-2 standard automated perimetry or OCT [36,46,47]. PERG amplitudes might disclose both RGC dysfunction and RGC loss. Therefore, the correlation between the presence of MvD and decreased PERG amplitude might suggest that eyes with MvD have a larger number of sick or dysfunctional RGCs. 

Quantitative analysis of MvD revealed that the angular extent of MvD was associated with the PERG N95 amplitude (*p* = 0.014). According to reports of the angular width of MvD [43,48,49], as the width gets broadened, the RNFL thinning and worsened MD were significantly detected. Similarly, robust linear relationships of the N95 amplitude with angular width, MD, average RNFL thickness, and average CD ratio were found in our study. Nonetheless, in multivariate analysis, angular width of MvD was the only significant factor associated with N95 amplitude. These findings denote a quantitative reflection of the compromised lesion of choroidal capillaries and dysfunctional RGCs. 

One report has recently demonstrated relationships between OCTA findings and PERG parameters [40]. Lee et al. [40] have reported that the N95 amplitude is correlated with average GCIPL thickness, peripapillary vessel density, and macular vessel density after adjusting for confounding factors. However, the N95 amplitude failed to show a significant relationship with peripapillary or macular superficial vessel density in our study. Such a discrepancy might be due to different inclusion criteria of patients. We limited patients to be those with glaucoma and PPA, while Lee et al. did not. In addition, different OCTA devices were used. In our study, DRI OCT Triton was used, while the study of Lee et al. used Cirrus-AngioPlex. Nonetheless, large-scale prospective studies are needed in the future to confirm these relationships. 

Regarding the relationship between the presence of MvD and PERG parameters, univariate analysis showed negative associations of the presence of MvD with P50 and N95 amplitudes. However, in multivariate analysis, only the N95 amplitude maintained its significant association. These findings are based on previous studies showing that the P50 amplitude stems from the soma of RGC, while the N95 amplitude originates from its axon [50,51]. Considering glaucoma development as a result of the failure of axoplasmic transport, it seems that the N95 amplitude would be more likely to represent changes in glaucoma than the P50 amplitude. Furthermore, different sources consisting of each amplitude could have led these findings. Namely, P50 amplitude is composed of 1/3 from preganglionic retinal sources and 2/3 rds from post-synaptic non-spiking activity within the ganglion cells, while the N95 amplitude is a reflection of the spiking activity of the cells [52]. Moreover, our precedent study [38] supports this finding as well. 

This study showed noticeable associations of the N95 amplitude with the presence of MvD and the angular width of MvD. However, it has several limitations. First, due to the nature of a cross-sectional study, we could not follow longitudinal changes in patients with MvD and reduced N95 amplitude to determine whether they truly experienced faster glaucoma progression than others. Moreover, changes in areas and angular widths were not followed. Several studies have shown that MvD is strongly associated with glaucoma progression [33,34,45,53,54]. Thus, a longitudinal and prospective study tracking changes in MvD and assessing their possible effects on PERG parameters would be needed to better understand the dynamics of glaucoma deterioration. Second, to estimate and evaluate MvD, the deep layer of the OCTA was used. During the process of the evaluation, there was a concern that retinal vessel signals evident on En Face might hinder the researchers’ ability to examine the deep layer precisely. However, recent studies have reported that the repeatability and reproducibility in measurement are good in the superficial layer and the deep layer [55,56]. Lastly, there are no established standard international reference ranges for PERG measurements. Nonetheless, in our previous study including normal eyes, the average N95 amplitude was 6.7 ± 1.6 (mean ± SD), and this could be used as a reference range [38]. However, the values obtained in this study are relative, and this requires additional caution when it comes to interpreting the data.

## 5. Conclusions

In summary, we investigated the probable impact of parapapillary deep-layer MvD on PERG parameters either qualitatively or quantitatively. The presence of MvD, and also its angular width, had robust associations with decreases of the N95 amplitudes. Since eyes with MvD would possess more dysfunctional RGCs than those without MvD, even if they have similar glaucoma severity, the presence of MvD could possibly mean that a fair number of RGCs are under mechanical or vascular stress in ONH. Our results indicate that more vigilance might be needed for glaucoma patients presenting with MvD.

## Figures and Tables

**Figure 1 jcm-11-02478-f001:**
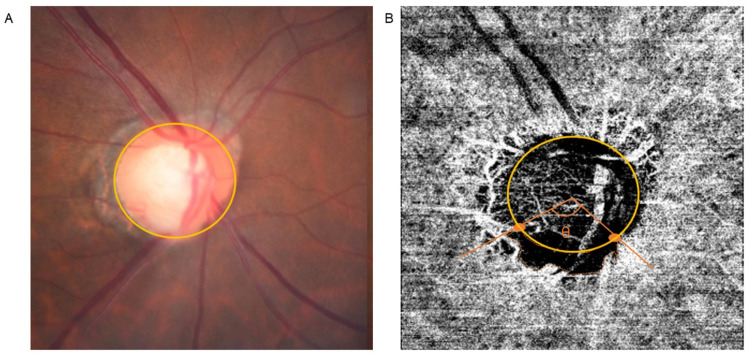
Measurement of the area and angular width of choroidal parapapillary microvasculature dropout (MvD). Fundus photo (**A**) and deep layer of OCT angiography (**B**) are shown. Yellow circle denotes margin of the optic disc. The area of MvD was defined by the demarcation of the orange line. As to the angular width of MvD, two lines were drawn from the center of the optic disc to the border points where MvD and the margins of the optic disc meet. The measured width between these two lines was defined as the angular width of MvD. All measurements were carried out by using a built-in calculating tool in the OCTA viewer software (IMAGEnet6; Topcon, Tokyo, Japan).

**Figure 2 jcm-11-02478-f002:**
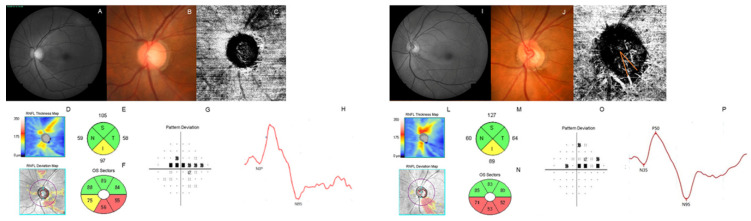
A representative case of comparing patients with MvD and without MvD. A female glaucoma patient (59 years old) without MvD (**C**) presented an inferotemporal RNFL defect (**A**,**B**) and showed 80 μm average RNFL thickness and 74 μm average GCIPL thickness (**D**–**F**). On her VF test, a central VF defect with MD of −2.35 dB was shown (**G**). The measured P50 and N95 amplitudes were 3.65 mV and 6.99 mV, respectively (**H**). Another female patient (60 years old) who had an inferotemporal RNFL defect (**I**,**J**) and MvD (**K**) showed a similar degree of glaucoma severity; MD was −1.95 dB (**O**), and average RNFL and GCIPL thicknesses were 85 μm and 70 μm, respectively (**L**–**N**). Nevertheless, the amplitudes of P50 and N95 in the patient with MvD were significantly lower (2.66 mV for P50 amplitude and 5.6 mV for N95 amplitude; (**P**)).

**Figure 3 jcm-11-02478-f003:**
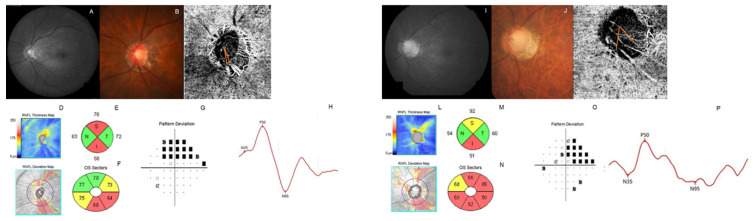
A representative case of comparing patients in terms of angular width of MvD. A 53-year-old male with multiple RNFL defects (**A**,**B**) had the MvD with an estimated angular width of 9.9° (**C**). Another patient (63 years old) (**I**,**J**) presented with larger MvD (angular width 54.9°; (**K**)). Regarding the level of disease severity, the patient with smaller MvD had average 67 μm RNFL and 72 μm GCIPL thicknesses (**D**–**F**), and VF MD was −8.43 dB (**G**). Similarly, the patient with larger MvD had average 64 μm RNFL and 65 μm GCIPL thicknesses and MD −8.92 dB (**L**–**O**). Despite the equivalent glaucoma stage, the patient with lager MvD saw significantly reduced P50 and N95 amplitudes than the patient with smaller MvD (2.46 mV and 2.96 mV, respectively, (**P**) vs. 2.58 mV and 6.58 mV, respectively, (**H**)).

**Table 1 jcm-11-02478-t001:** Patient’s Demographics (N, 79).

	Mean ± SD
Age (Year)	54.33 ± 12.27
Sex (Male, %)	46 (47.9%)
HTN (Yes, %)	14 (16.9%)
DM (Yes, %)	4 (4.8%)
CCT (μm)	540.37 ± 39.85
AL (mm)	24.98 ± 1.25
Pattern Electroretinography	
N35 implicit time (ms)	22.75 ± 5.58
P50 implicit time (ms)	48.31 ± 4.29
N95 implicit time (ms)	101.72 ± 10.04
P50 amplitude (mV)	2.70 ± 0.99
N95 amplitude (mV)	4.67 ± 1.61
Visual Field	
MD (dB)	−4.64 ± 6.53
PSD (dB)	4.97 ± 3.50
OCT	
Average RNFLT (μm)	74.51 ± 13.23
Rim Area	0.86 ± 0.22
Disc Area	1.97 ± 0.46
Average CD ratio	0.72 ± 0.11
Average GCIPLT (μm)	69.58 ± 8.76
OCT Angiography	
Signal Strength	65.92 ± 5.91
Superficial VD	39.36 ± 4.68
Macular superficial VD	34.26 ± 2.71
Total PPA area (mm^2^)	2.78 ± 4.64
PPA VD	49.77 ± 12.26
Area of MvD (mm^2^)	0.38 ± 0.40 *
PPA area except MvD (mm^2^)	2.45 ± 4.60 *
Angle of MvD (°)	52.21 ± 31.69 *
Angle of RNFLD (°)	60.95 ± 56.68 *

SD, standard deviation; HTN, hypertension; DM, diabetes mellitus; CCT, central corneal thickness; AL, axial length; MD, mean deviation; PSD, pattern standard deviation; OCT, optical coherence tomography; RNFLT, retinal nerve fiber layer thickness; CD, cup disc; GCIPLT, ganglion cell inner plexiform layer thickness; VD, vessel density; PPA, parapapillary atrophy; MvD, microvascular dropout; RNFLD, retinal nerve fiber layer defect. ***** Values were measured in eyes with MvD.

**Table 2 jcm-11-02478-t002:** Comparison of Patient’s Characteristics in terms of Presence of Circumparapapillary Choroidal Microvasculature Dropout.

		MvD (−) (N, 33)	MvD (+) (N, 46)	*p* Value
Age (year)		53.39 ± 13.01	55.93 ± 12.98	0.394 *
Sex (male, %)		15 (45.5%)	23 (50%)	0.820 ^#^
CCT (μm)		531.79 ± 43.96	541.59 ± 32.65	0.275 *
AL (mm)		25.08 ± 1.32	25.37 ± 1.04	0.346 *
Pattern Electroretinography				
N35 implicit time (ms)		22.20 ± 5.71	23.48 ± 5.59	0.322 *
P50 implicit time (ms)		48.38 ± 4.38	48.57 ± 4.58	0.859 *
N95 implicit time (ms)		100.89 ± 9.12	103.00 ± 11.45	0.383 *
P50 amplitude (mV)		3.03 ± 1.10	2.42 ± 0.87	**0.007** *
N95 amplitude (mV)		5.24 ± 1.76	4.14 ± 1.53	**0.004** *
Visual Field				
MD (dB)		−3.67 ± 5.20	−6.09 ± 7.92	0.107 *
PSD (dB)		4.52 ± 3.55	5.37 ± 3.57	0.300 *
OCT				
Optic disc parameter	Disc Area (mm^2^)	1.92 ± 0.39	1.96 ± 0.52	0.729 *
	Rim Area (mm^2^)	0.85 ± 0.17	0.82 ± 0.23	0.532 *
	Average CD ratio	0.72 ± 0.12	0.73 ± 0.12	0.600 *
Average RNFLT (μm)		75.06 ± 13.06	72.02 ± 13.25	0.319 *
Average GCIPLT (μm)		70.52 ± 8.70	67.82 ± 7.59	0.171 *
OCT Angiography				
Signal Strength		65.09 ± 6.73	65.37 ± 5.75	0.844 *
Parapapillary VD		38.89 ± 5.09	39.54 ± 4.13	0.565 *
Macular VD		33.63 ± 2.74	34.77 ± 2.65	0.193 *

MvD, microvascular dropout; CCT, central corneal thickness; AL, axial length; MD, mean deviation; PSD, pattern standard deviation; OCT, optical coherence tomography; CD, cup disc; RNFLT, retinal nerve fiber layer thickness; GCIPLT, ganglion cell inner plexiform layer thickness; VD, vessel density. Factors with statistical significance are shown in bold. Data are mean ± standard deviation unless otherwise indicated. * Student *t*-test. ^#^ Chi-Square test.

**Table 3 jcm-11-02478-t003:** Logistic Regression Analysis of Factors associated with the Presence of Circumparapapillary Choroidal Microvasculature Dropout.

	Univariate		Multivariate	
	β (95% CI)	*p* Value	β (95% CI)	*p* Value
Age (year)	0.015 (0.980–1.051)	0.416		
MD (dB)	−0.057 (0.873–1.023)	0.164		
PSD (dB)	0.075 (0.943–1.232)	0.274		
N35 implicit time (ms)	0.038 (0.958–1.126)	0.354		
P50 implicit time (ms)	0.018 (0.919–1.128)	0.726		
N95 implicit time (ms)	0.017 (0.973–1.063)	0.451		
P50 amplitude (mV)	−0.620 (0.324–0.894)	**0.017**		
N95 amplitude (mV)	−0.390 (0.502–0.913)	**0.011**	−0.668 (0.296–0.887)	**0.017**
Average RNFLT (μm)	−0.020 (0.946–1.015)	0.260		
Average CD ratio	1.311 (0.083–165.9)	0.499		
Average GCIPLT (μm)	−0.041 (0.904–1.020)	0.185		
Peripapillary VD	0.029 (0.927–1.144)	0.583		
Macular VD	0.163 (0.974–1.423)	0.091		

CI, confidence interval; MD, mean deviation; PSD, pattern standard deviation; RNFLT, retinal nerve fiber layer thickness; CD, cup disc; GCIPLT, ganglion cell inner plexiform layer thickness, VD, vessel density. Variables with *p* < 0.10 were included in the multivariate analysis. Factors with statistical significance are shown in bold.

**Table 4 jcm-11-02478-t004:** Linear Regression Analysis evaluating the Factors associated with Circumparapillary Choroidal Microvasculature Dropout Angular Width in the subgroups of Eyes with Choroidal Microvasculature Dropout.

	Univariate		Multivariate	
	B	*p* Value	B	*p* Value
Age (year)	0.354	0.347		
MD (dB)	−1.371	**0.023**		
PSD (dB)	1.780	0.191		
N35 implicit time (ms)	0.869	0.319		
P50 implicit time (ms)	−0.803	0.452		
N95 implicit time (ms)	0.029	0.946		
P50 amplitude (mV)	−8.164	0.141		
N95 amplitude (mV)	−7.612	**0.014**	−7.612	**0.014**
Average RNFLT (μm)	−0.796	**0.027**		
Average CD ratio	73.736	0.065		
Average GCIPLT (μm)	−0.965	0.188		

MD, mean deviation; PSD, patten standard deviation; RNFLT, retinal nerve fiber layer thickness; CD, cup disc; GCIPLT, ganglion cell inner plexiform layer thickness. Variables with *p* < 0.10 were included in the multivariate analysis. Factors with statistical significance are shown in bold.

**Table 5 jcm-11-02478-t005:** Linear Regression Analysis evaluating the Factors associated with the Pattern Electroretinogram Amplitudes.

	P50 Amplitude				N95 Amplitude			
	Univariate		Multivariate		Univariate		Multivariate	
	β	*p* Value	β	*p* Value	β	*p* Value	β	*p* Value
Age (year)	−0.028	**0.001**	−0.026	**0.005**	−0.049	**<0.001**	−0.047	**0.001**
MD (dB)	0.025	0.106			0.095	**<0.001**		
PSD (dB)	0.003	0.921			−0.093	**0.052**		
Average RNFLT (μm)	0.013	**0.093**			0.046	**<0.001**		
Average CD ratio	−1.231	0.171			−2.629	**0.069**		
Average GCIPLT (μm)	0.036	**0.003**	0.038	**0.007**	0.086	**<0.001**	0.067	**0.003**
Presence of MvD	−0.617	**0.007**	−0.428	0.051	−1.090	**0.004**	−0.815	**0.020**
Superficial VD	−0.015	0.553			−0.007	0.854		
Macular superficial VD	−0.034	0.421			−0.018	0.782		

MD, mean deviation; PSD, patten standard deviation; RNFLT, retinal nerve fiber layer thickness; CD, cup disc; GCIPLT, ganglion cell inner plexiform layer thickness; MvD, microvascular dropout; VD, vessel density. Variables with *p* < 0.10 were included in the multivariate analysis. Factors with statistical significance are shown in bold.

## Data Availability

The data presented in this study are available on request from the corresponding author. The data are not publicly available due to limited access to electronic medical record of Seoul St.Mary’s Hospital.

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
