# Peer review of "Attenuated Amplitude of Pattern Electroretinogram in Glaucoma Patients with Choroidal Parapapillary Microvasculature Dropout"

_jcm, 2022, doi:10.3390/jcm11092478_

Round 1
Reviewer 1 Report
I would like to thank the authors for this precise and well-commented work.
It brings only few criticisms or remarks.
1° In the introduction, more emphasis should be placed on the fact that pERG is considered a good test for detecting glaucoma. In this sense, some sentences are on the edge of the truth.
You say "The strength of pERG leads us to assume that glaucomatous eyes with attenuated pERG parameters might have a substantial sick RGCs which could increase under substantial amounts of mechanical and vascular stress". But this is widely accepted. Similarly in the limitations, it is written that "Second, we only included glaucoma patients presenting PPA. Therefore, it would be hard to apply findings of this study to general open angle glaucoma patients." Again, this is an accepted fact.
2° Figure 1, the MvD calculation method is not clear. Why does it stop in the middle of an avascular zone?
3° Table 1, it would be interesting to give normal value when applicable for readers who are not ophthalmologists or not familiar with pERG
4° Table 3 and 4, two values ​​are indicated in bold as if they were significant when they are not.
Reviewer 2 Report
The article is original and attractive
Line 64: Exclusion criteria; were myopic eyes excluded? If yes, by which degree of myopia?
Line 88: Which RNFL defects? Please specify
Line 228: "paltrier" (??)
In the discussion (line 303), it should be pointed out that the less significant P50 wave decrease in N95 may be explicable by its predominantly macular genesis compared with that from ganglion cells in N95
